# Effectiveness and cost-effectiveness randomised controlled trial of basic versus biofeedback-mediated intensive pelvic floor muscle training for female stress or mixed urinary incontinence: protocol for the OPAL (optimising pelvic floor exercises to achieve long-term benefits) trial mixed methods longitudinal qualitative case study and process evaluation

Aileen Grant,[1] Sarah Dean,[2] Jean Hay-Smith,[3] Suzanne Hagen,[4] Doreen McClurg,[4] Anne Taylor,[5] Marija Kovandzic,[5] Carol Bugge[5]

► http://dx.doi.org/10.1136/bmjopen-2018-024153

For numbered affiliations see end of article.

**Correspondence to**
Dr Aileen Grant;
a.grant17@rgu.ac.uk

## ABSTRACT

**Introduction** Female urinary incontinence (UI) is common affecting up to 45% of women. Pelvic floor muscle training (PFMT) is the first-line treatment but there is uncertainty whether intensive PFMT is better than basic PFMT for long-term symptomatic improvement. It is also unclear which factors influence women's ability to perform PFMT long term and whether this has impacts on long-term outcomes. OPAL (optimising PFMT to achieve long-term benefits) trial examines the effectiveness and cost-effectiveness of basic PFMT versus biofeedback-mediated PFMT and this evaluation explores women's experiences of treatment and the factors which influence effectiveness. This will provide data aiding interpretation of the trial findings; make recommendations for optimising the treatment protocol; support implementation in practice; and address gaps in the literature around long-term adherence to PFMT for women with stress or mixed UI.

**Methods and analysis** This evaluation comprises a longitudinal qualitative case study and process evaluation (PE). The case study aims to explore women's experiences of treatment and adherence and the PE will explore factors influencing intervention effectiveness. The case study has a two-tailed design and will recruit 40 women, 20 from each trial group; they will be interviewed four times over 2 years. Process data will be collected from women through questionnaires at four time-points, from health professionals through checklists and interviews and by sampling 100 audio recordings of appointments. Qualitative analysis will use case study methodology (qualitative study) and

the framework technique (PE) and will interrogate for similarities and differences between the trial groups regarding barriers and facilitators to adherence. Process data analyses will examine fidelity, engagement and mediating factors using descriptive and interpretative statistics.

**Ethics and dissemination** Approval from West of Scotland Research Ethics Committee 4 (16/LO/0990).

## Strengths and limitations of this study

► The strength is the design of an in-depth, pre-planned and theoretically driven process evaluation (PE) and longitudinal comparative qualitative case study to support understanding of two complex interventions.
► This evaluation will sample from women from both intervention arms and professionals from all centres delivering the interventions.
► The strength of the case study is the large number of cases and its comparative longitudinal design that works in parallel with the trial to explore implementation, context and adherence to pelvic floor muscle training treatment regimes.
► The strength of the PE is the range of data for triangulation.
► The limitation of this work is that we did not collect any data from sites prior to them delivering the interventions.

Findings will be published in journals, disseminated at conferences and through the final report.
**Trial registration number** ISRCTN57746448.

## INTRODUCTION

Urinary incontinence (UI), the involuntary loss of urine, is a common condition in women. The prevalence of UI depends on the definition, using a broad definition between 5% and 69% is reported but most studies report the range 25%–45% of women internationally.[1] The main types of UI are stress, urgency and mixed incontinence, with stress incontinence being the most prevalent affecting approximately 50% of women with UI; mixed UI affects 7.5%–25% and urgency affects approximately 1%–7% of women with UI. The cost to the UK's National Health Service (NHS) was estimated at £233 million in 2000[2] and the personal costs to women estimated to be £178 million.[3]

Pelvic floor muscle training (PFMT) is the first line treatment for stress and mixed UI[4] and there is good evidence to show PFMT is effective in the short term.[5 6] However, it is not clear whether or how common intensifiers of PFMT (eg, more contact with health professionals, use of adjuncts such as biofeedback) increase longer-term PFMT adherence and the duration of effect. A recent series summarising the PFMT adherence literature[7–11] does suggest that self-efficacy is one factor influencing women's ability to continue to perform self-care, such as PFMT, after treatment but there is a need for more research in this area to identify the full range of factors and how these impact on their UI symptoms in the longer term.

Biofeedback (audio and/or visual feedback from a pressure or force-sensitive or electromyography device of a contraction of pelvic floor muscles) is often used as an adjunct to PFMT. A Cochrane review of the effectiveness of PFMT augmented with biofeedback concluded that women receiving biofeedback were more likely to report improved symptoms or cure than women who did not receive biofeedback; however, the effect might be confounded by greater amounts of health professional contact in women receiving biofeedback.[12] As a result, Hagen and colleagues are conducting a large multicentre randomised control trial of the effectiveness and cost-effectiveness of long-term adherence to basic versus biofeedback-mediated intensive PFMT for female stress or mixed UI (OPAL), with the same amount of health professional contact in both trial groups.[13] Unique to the trials conducted so far in this field, the OPAL trial has an embedded mixed-methods process evaluation (PE) to explore factors impacting on short-term and long-term adherence to PFMT, and any other mediating factors for treatment delivery and effectiveness. This PE will draw on analysis of multiple datasets generated throughout the trial including the interviews with the therapists, audio recorded consultations, treatment checklists and trial participant questionnaires, as well as on the findings from the longitudinal qualitative interviews exploring women's experiences of the OPAL intervention.

### The OPAL trial

The OPAL trial is a large multicentre, pragmatic randomised controlled trial of two active treatment interventions, basic PFMT and biofeedback-mediated PFMT for the treatment of stress or mixed UI. The trial has been fully described in the companion trial protocol paper and further information is available on the trial website (www.opaltrial.co.uk). In brief, 'Intensive' PFMT consists of basic PFMT with adjunctive biofeedback at every clinic appointment and at home. Both 'basic' and 'intensive' treatment groups are offered six clinic appointments over the 16-week intervention period. Women in both groups receive pelvic floor muscle assessment,[14] vaginal palpation, verbal feedback and individualised exercise prescription based on this assessment, advice for bowel and bladder symptoms and tailored lifestyle advice. Treatment is delivered by women's health physiotherapists or continence nurses who have received training designed and delivered by the trial team.[13] The premise is that biofeedback will intensify PFMT by offering women more visible information about exercise outcomes, support motivation and enhance behavioural skill (eg, improved contraction technique and confidence). The anticipated outcome of adjunctive biofeedback is improved quality and quantity of exercise, with the expectation that this will increase treatment benefit and reduce symptoms. However, the biofeedback intervention can only work to intensify PFMT if women use the device.

### The OPAL longitudinal qualitative case study and PE

The OPAL evaluation has a comprehensive and well-resourced longitudinal qualitative comparative case study and mixed-method PE running parallel to the trial to provide important insights into the trial's processes and outcomes: to explain how the interventions worked (what factors led to the observed effect); assist in interpretation of the trial findings; and facilitate transferability into clinical practice (if effective) and further research (in particular if not effective). This design is rather unusual but allows the researchers to preserve multiple perspectives and contribute to the robustness of the complex analysis of factors mediating the trial delivery and findings. These studies will be cross-referenced to support more comprehensive explanations of how the interventions work and assist in interpreting the trial findings. We describe each study in more detail below:

The qualitative case study is longitudinal in nature to mirror the trial data collection, to capture the dynamic nature of treatment delivery and participants' experiences, to generate more in-depth data about what works well (or not) when intensifying treatment regimens, and most importantly, in this trial to better understand the factors important to long-term adherence to PFMT. Within the trials literature, this longitudinal methodological design is rare, but it is needed to capture the

dynamic nature of implementation and adherence to long-term PFMT treatment regimens; furthermore, the commissioning call for this work explicitly requested a qualitative study to explore women's experience and barriers to adherence. This qualitative case study will also make an important contribution to understanding the behavioural aspects of therapy which are rarely documented.

The PE aims to provide transparency by exploring the implementation of the interventions in context.[15] Variation in implementation of the OPAL interventions is likely to be due to: the different intervention components; the diverse clinical settings (university hospital, district general hospital, community settings) and health professionals delivering (women's health physiotherapists and incontinence specialist nurses); and, the contextual and personal differences between women receiving the interventions and adhering to PFMT long-term. These different factors are likely to influence how well the interventions are integrated within existing practice and women's lives and may impact on the trial findings. As a result, we believe it is important to explore delivery (including fidelity and dose—if the intervention components are being delivered as intended and in what quantity across the different trial centres), response (participant's experiences of the intervention) and maintenance (how and why these processes are sustained over time or not by women and therapists during the intervention delivery and by women after the intervention delivery is complete).[15] For delivery, we are particularly interested in fidelity of function[16] as clinicians were given a protocol to follow with the expectation that they would individualise this (the exact exercise prescription, lifestyle advice and so on) to the care each woman required; however, there were also specific behaviour change techniques that were to be used for all women (and not individualised) and so we will also be investigating fidelity of form for this element of the interventions. Women's experiences of receiving treatment in the clinic and how they engage with PFMT at home is important to how they respond to the interventions. As OPAL is interested in long-term adherence to PFMT, it is important to explore how women maintain their adherence (or not) and the factors which facilitated or impeded this. It is hypothesised that self-efficacy will be a mediating factor to intervention effectiveness. The longitudinal qualitative case study will support the PE by providing data on 'response' and 'maintenance' to explain the women's experiences of receiving treatment, how and why they engaged with PFMT and/or biofeedback PFMT in the clinic and at home during treatment and how and why they maintained engagement (or not) over time.

### Aim and objectives
The overall aim of this mixed method evaluation is to understand why the interventions were effective or ineffective, understand implementation and the issues in short-term and long-term adherence to PFMT and to inform wider implementation (or further research if ineffective) as well as the PFMT intervention literature. Although interlinked, we describe each parallel evaluation (see figure 1) separately to facilitate clarity as each evaluation has a specific objective.

### Longitudinal comparative qualitative case study objectives
The objectives were to investigate women's experiences of the interventions, both basic and intensive PFMT, to identify the barriers and facilitators which impact on adherence in the short-term and long-term, to explain the process through which they influence adherence and to identify whether these differ between randomised groups.

### PE objective
The objectives were to identify and investigate the possible mediating factors that impact the effectiveness of the intervention (including intervention fidelity), how these mediating factors influence effectiveness and whether the factors differ between randomised groups.

### Management and governance
The trial was registered on the ISRCTN registry (ISRCTN 57746448).

The longitudinal comparative qualitative case study and PE management group has a mix of relevant clinical, qualitative, quantitative and theoretical skills and experience[17] and meets regularly to discuss the research management and emerging findings. In order to ensure allocation concealment is maintained, it has been agreed that this team will not discuss or present findings from this study with any staff from the main trial project management group. As a result, these meetings are closed.

## METHODS AND ANALYSIS
### Methods for longitudinal comparative qualitative case study
A longitudinal, qualitative, two-tailed case study design[18] utilising semistructured interviews with purposively sampled women from both groups to explore the experience of, and adherence to, the trial interventions. The tails will be the experimental and comparator groups of the trial. Using a two-tailed case study design complements the trial design in its comparative focus. The analysis, like the trial analysis, is set up to explore differences between the 'tails' or groups. In the case study design, the differences can be exposed and support understanding of effectiveness (or non-effectiveness).

### Sampling and recruitment strategy
Thirty to 40 randomised women (15 to 20 in each tail) will be purposively sampled for variance in centre type (district general hospital, university hospital, community delivered service), women's type of UI (stress or mixed) and therapist type (physiotherapist/nurse), as case studies. Purposively selected women, who have consented

# THE OPAL TRIAL

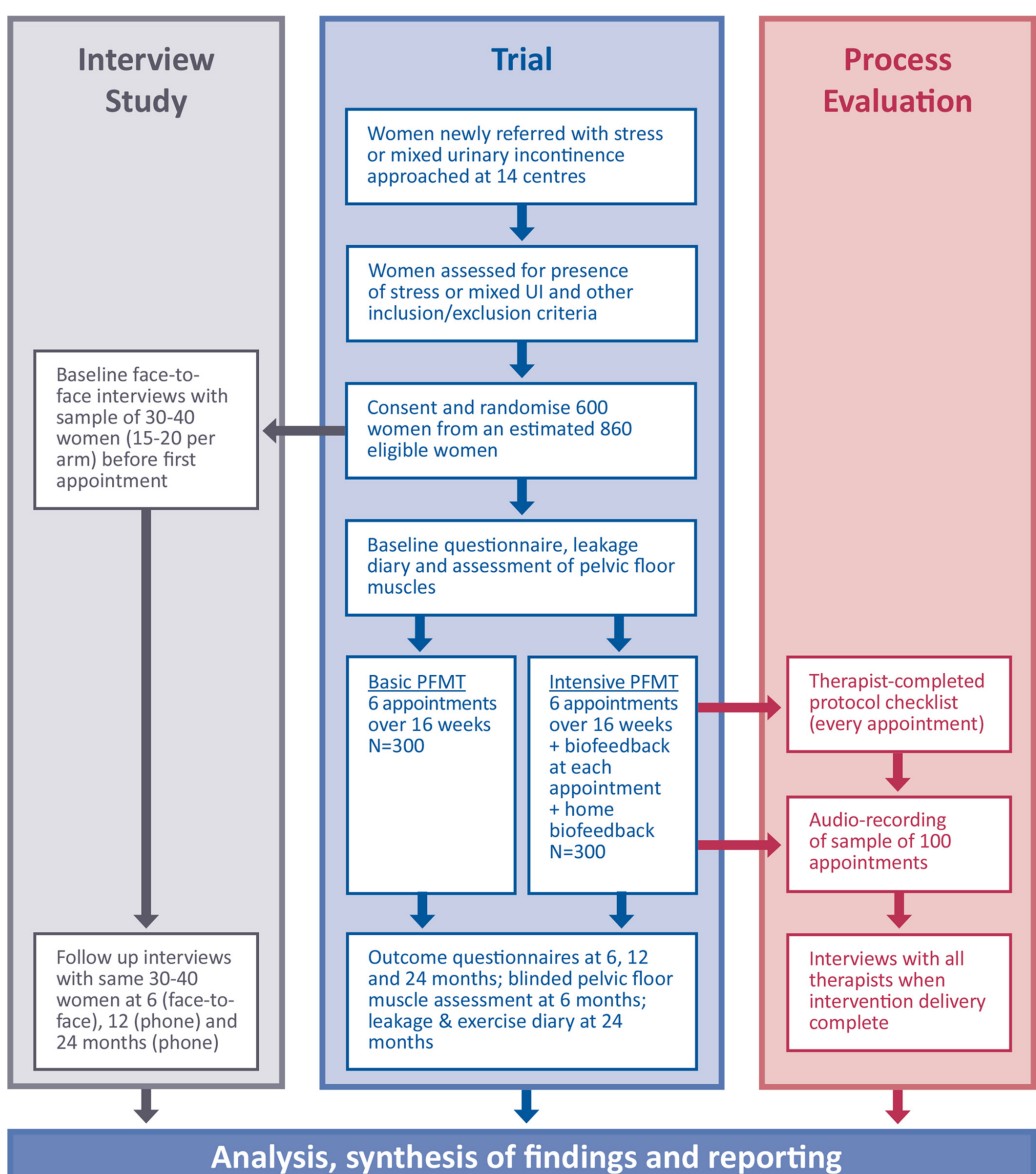

**Figure 1**  Trial flow diagram. PFMT, pelvic floor muscle training.

to the trial, will be given an additional information leaflet and asked if they are interested in taking part in an interview study. Interested women will be contacted by telephone approximately a week later to ask if they would like to participate. If a woman declines to consent to the interview study, another woman with similar characteristics will be selected and approached. Formal written consent will be collected at the time of the first face-to-face interview. We anticipate some attrition of women by the 12-month and 24-month interviews, however, we have oversampled to ensure sufficient heterogeneity to identify barriers and facilitators to respond to the interventions, and to generate hypotheses about the factors associated with differing outcomes, including long-term adherence to PFMT.

## Qualitative data collection

Data will be collected by a series of semistructured interviews at baseline, 6, 12 and 24 months (12 and 24 months by telephone). Each interview will have a specific focus:

► Baseline (pretreatment) interview (face-to-face) will explore the woman's experience of UI, the social contexts within which she experiences UI and her expectations of treatment.

► Six-month interview (face-to-face) will explore the woman's experience of the trial intervention, her adherence to therapy appointments and the prescribed programme, and factors that affected that adherence and her perceptions of treatment outcome.

► 12-month interview (telephone) will explore the woman's experience of UI postintervention, of the

intervention, of factors that influence ongoing adherence to PFMT and of treatment outcome.

► 24-month interview (telephone) will explore the same issues as at 12 months but with a focus on the longer term.

Interview data will be collected using a password protected audio digital recorder.

## Qualitative case study data analysis

Interview audio recordings will be transcribed verbatim and entered into QSR NVivo software to support analysis. The analysis will be iterative with data generation and will take place on four interacting levels to facilitate within and cross-case comparisons to ensure comprehensive examination which exposes the similarities and differences between the experimental and comparator groups and identifies barriers and facilitators to adherence:

► *At the level of the individual interview.* An initial *a priori* coding scheme will be applied that focuses on core areas of interest: specifically, women's experiences of UI; experience of PFMT +/−biofeedback; factors that influence adherence to supervised treatment and home exercise; and perceptions of treatment outcome. Research team discussions and constant iterative coding will further develop the coding scheme. The combination of the *a priori* scheme and inductive codes will aim, at this stage, to identify barriers and facilitators that influence adherence.

► *At the level of the case (woman).* Case summaries will be written with a focus on creating an understanding of women's experience in our areas of interest: the problem, the treatment, adherence to supervised treatment and home exercise; perceptions of treatment outcome and how these factors interact. Analysis at this stage will focus on identifying issues relating to changes over time and in developing theoretical propositions to guide subsequent analysis (Yin 2003).

► *At the level of the trial group.* All the cases for one trial group will be collected together and consistencies/inconsistencies searched for. The aim of analysis at this stage is to identify the core barrier and facilitators within the trial group, the detailed explanations for them and interactions between them.

► *At the tail level.* The experimental and comparator tails will be compared with one another using the theoretical propositions. The aim of the analysis at this point is to identify similarities and differences in barriers and facilitators between the trial groups.

## Methods for PE

This mixed methods PE will focus on assessing the fidelity of function (with additional checking of fidelity of form for the behavioural change techniques)[16] and explore delivery and implementation with therapists delivering the interventions in all centres. Fidelity will be assessed using theory-based and protocol-linked checklists completed by therapists at each appointment a woman attends and audio recordings of a purposive sample of 100 therapy appointments which will be assessed against a theory-driven framework derived from the trial protocol. Qualitative telephone interviews will be conducted with health professionals delivering the interventions at the end of their involvement in the trial. This is a concurrent study design with all methods being used simultaneously.

## PE sampling and recruitment strategy and data collection

1. A protocol checklist will be completed by all therapists after each appointment. The checklist will allow assessment of protocol deviations (as defined in the main trial and for statistical analysis purposes). It will also allow explanation of the clinically informed and permitted variance in delivery through tailoring of the intervention based on individual women's needs; alongside an explanation of the reasons, the variance occurred.

2. Audio recordings will be conducted with a purposive sample of 100 appointments. These will be sampled for heterogeneity across: the comparator and experimental groups; the different therapy appointments (in both groups, six therapy appointments across 16 weeks); centre (aiming for inclusion of appointments at all centres); woman's type and severity of UI; and therapist type (physiotherapist or incontinence nurse). Each centre will be provided with a password-protected audio digital recorder to record the selected appointments. We will aim to oversample audio recordings for the first and last appointments due to our a priori hypothesis that treatment delivery may be more intensive and concentrated in the first and last appointments. The first appointment is a longer consultation and involves important education and training for being taught how to properly perform PFMT, where relevant including the teaching of biofeedback, and for understanding any changes to lifestyle that may be recommended. The last appointment is important because therapy is coming to an end and women are being given important information to allow them to self-care, such as instructions regarding the maintenance dose of PFMT. If participants have signed on their consent form that they are willing to have an appointment audio recorded, the researcher will purposively select participants and telephone them to ask if they would be willing to have a specific appointment recorded, answer any questions they may have and to take verbal consent. Therapists will then be informed which participants and appointments to record. The audio recording devices will be returned to the researcher at regular intervals to download the data and to ensure the audio consultations are transcribed verbatim.

3. Semistructured telephone interviews will be undertaken with therapists who have been involved in delivering the interventions at the end of their participation. We aim to interview at least one therapist from each site. A topic guide developed from the literature and from issues which have arisen during delivery of the intervention and from patient interviews will be utilised. The interviews will explore the therapist's experiences

of delivering the PFMT and BF PFMT interventions, including their perspectives on adherence to delivering the protocols and women's adherence to the intervention.

4. The trial follow-up questionnaires will include questions on women's adherence and self-efficacy to PFMT during and after the supervised intervention which will contribute to the PE. These data will be analysed like secondary outcome measures (please see Hagen *et al* for more details).[13] In addition, the interview study (described earlier) will provide data from the women about adherence to treatment to feed into the PE.

## PE analysis

Each data source will be analysed individually in the first instance to reach separate conclusions:

1. Data from the checklists completed by the therapists will be summarised descriptively to report the extent to which there were consistent with the protocol. Free-text comments from the therapists relating to any barriers and facilitators to delivering the protocol and any variations due to individual tailoring they experienced during appointments will be coded using a coding framework developed using content analysis with a 10% representative sample of appointments.

2. Data from interviews with therapists will be analysed using the Framework Approach (Ritchie *et al*).[19] Following familiarisation with the data, a thematic framework will be developed and applied across the data set. Data will then be tabulated and conceptual maps used to make links between themes. Where possible in vivo codes and concepts will be utilised.

3. A quantitative coding scheme will be developed for analysis of the appointment audio recordings. This coding scheme will be developed using the intervention protocols and the theory underlying the protocols, and data generated from a purposive sample of recordings (treatment arm, visit number and site). The coding scheme will contain explicit guidance as to what codes have to be applied in what circumstances. The audio recordings will then be coded and entered into SPSS. Coded data will then be subject to descriptive and interpretive analyses.

4. Data from the main study self-report questionnaires will be analysed in line with the Statistical Analysis Plan. Specifically, and as indicated in the main standard operating procedure, we will undertake a mediational analysis as part of the quantitative PE, and this will focus on the self-efficacy data as our hypothesised mediating variable. The details of the mediational analyses are described in the PE analysis plan.

## Synthesising the data from the longitudinal comparative case study and the PE

Synthesis will be undertaken once both analyses are complete, where the case study analysis will be cross-referenced and synthesised with the PE findings. Data synthesis will be undertaken whereby the findings from individual data sources will be presented in matrices that bring together key issues from the different analyses to draw hypotheses about why the intervention components were implemented more successfully than others and explore if there is a synergistic effect and if these are related to patient's experiences of BF PFMT and PFMT treatment. We will also elucidate the causal mechanisms which lead to change (or not) in each arm and suggest which components of the interventions were more successful and why. The combined data will also allow us to present a nuanced analysis of context. This will provide important information about the implementation of the interventions and if and how the interventions can be transferred to other settings.

## Integration of the case study, PE and trial findings

Both studies will be analysed separately before the outcome of the trial is known and the main trial will be analysed without knowledge of the longitudinal case study and PE findings.[20] The main trial research team and the research team have agreed that no case study/PE data will be shared before the trial code is broken. Following the identification of the main trial findings, subsequent analysis of the case study and PE data will be undertaken. There will be transparency in reporting which findings were identified prior to, and after, the trial findings were known.

## Public patient involvement

OPAL has a woman who has experience of UI as a grant holder and has patient representation on the Trial Steering Committee. Through these inputs, the research question, outcome measures and study design have all had women's perspectives included. Patient-facing recruitment materials were all reviewed by women with UI. Women were not involved in other aspects or study recruitment or conduct. Participants will all receive a lay summary of the results of the study. The burden of the intervention was assessed by our grant holder with UI. Our grant holder is named as an author on our main trial protocol paper.

## ETHICS AND DISSEMINATION

The study will be conducted in accordance with the ICH GCP Note for Guidance on Good Clinical Practice. Favourable ethics opinion covering recruitment across all UK NHS centres was obtained and approved from the Regional NHS Ethics Committee and local R&D departments.

Final trial results will be disseminated to the funding body, the NIHR Health Technology Assessment Programme. The trial results will be then submitted to peer-reviewed journals and presented at international conferences. Participants will be provided with a summary of the results.

## Study timelines

Funding for this study was approved on the 14 August 2012.

The Trial and qualitative evaluation and PE all started on 1 September 2013 and finished on 30 November 2018.

**Author affiliations**
[1]School of Nursing and Midwifery, Robert Gordon University, Aberdeen, UK
[2]South West Peninsula Collaboration for Leadership in Applied Research in Health Care (PenCLAHRC), University of Exeter, Exeter, UK
[3]Rehabilitation Teaching and Research Unit, University of Otago, Wellington, New Zealand
[4]Nursing Midwifery and Allied Health Professions Research Unit, Glasgow Caledonian University, Glasgow, UK
[5]Faculty of Health Sciences and Sport, University of Stirling, Stirling, UK

**Acknowledgements** The authors also acknowledge the support of the National Institute for Health Research, through the Comprehensive Clinical Research Network. SD's position is partly supported by the National Institute for Health Research (NIHR) Collaboration for Leadership in Applied Health Research and Care South West Peninsula at the Royal Devon and Exeter NHS Foundation Trust.

**Contributors** CB, SGD, JH-S: involved in the initial conceptualisation and study design. AG and AT: involved with participants and data collection. CB, SGD, AG, JH-S, MK and AT: involved in data analysis. SH: is the PI and along with DMcC advise on interpretation of the data in light of the trial results. AG: wrote the paper. All authors contributed to, read and approved the final manuscript. CB: is responsible for this manuscript.

**Funding** This project, Optimising Pelvic floor Exercises to Achieve Long-term benefits (OPAL) is funded by the NIHR Health Technology Assessment Programme (project reference: 11/71/03).

**Disclaimer** The views expressed are those of the authors and not necessarily those of the NHS, the NIHR or the Department of Health and Social Care.

**Competing interests** None declared.

**Patient consent for publication** Not required.

**Ethics approval** The West of Scotland Research Ethics Committee 4 (6/LO/0990).

**Provenance and peer review** Not commissioned; externally peer reviewed.

**Data sharing statement** Technical appendix, statistical code, and datasets (subject to agreement with the TSC and funders) will be made available at the end of the trial.

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
