## [Reviewer comments · BMJ Open]

This paper was submitted to a another journal from BMJ but declined for publication following peer review. The authors addressed the reviewers' comments and submitted the revised paper to BMJ Open. The paper was subsequently accepted for publication at BMJ Open.

(This paper received three reviews from its previous journal but only two reviewers agreed to published their review.)

ARTICLE DETAILS

TITLE (PROVISIONAL)	Effectiveness and cost-effectiveness randomised controlled trial of basic versus biofeedback-mediated intensive pelvic floor muscle training for female stress or mixed urinary incontinence: protocol for the OPAL (Optimising Pelvic floor Exercises to Achieve Long-term benefits) trial mixed methods longitudinal qualitative case study and process evaluation
AUTHORS	Grant, Aileen; Dean, Sarah Gerard; Hay-smith, Jean; Hagen, Suzanne; McClurg, Doreen; Taylor, Anne; Kovandzic, Marija; Bugge, Carol

VERSION 1 – REVIEW

REVIEWER	Hege Hølmo Johannessen Østfold Hospital Trust Norway
REVIEW RETURNED	14-Jun-2018

GENERAL COMMENTS	I welcome this study and applaud the authors for conducting a well planned and long awaited trial about the effectiveness of individual PFMT, long term treatment adherence and how to optimize treatment. The protocol is well written and easy to follow. However, I do miss an overview of when the study was initiated/ data collection started, in both the original intervention trial, but also in the mixed methods trial, and the expected time of finalizing data collection. The authors have included questionnaires about bowel health and pelvic organ prolapse which is I believe to be an important aspect. Thus, I would also like to enquire about whether the authors plan to include any questions in the planned interviews about other pelvic floor disorders (PFD), and how experiencing multiple PFDs may influence adherence and the participants' experience of PFMT and the individual follow up.
---

REVIEWER	Kate Stephen University of the Highlands & Islands, UK
REVIEW RETURNED	16-Jul-2018

GENERAL COMMENTS	Tables in pages 34/40 and 35/40 did not appear correctly (went off page and over text). Spirit Checklist 32 A model consent form does not seem to have been included as an appendix
---

REVIEWER	Thomas Gray Sheffield Teaching Hospitals NHS Foundation Trust, UK
REVIEW RETURNED	31-Oct-2018

GENERAL COMMENTS	This is an excellent protocol with clearly defined methodology and clear questions and outcomes. My only observation is that there may also be an opportunity to administer some validated patient reported outcomes which assess experiences with treatment, these could add some quantitative data to this strand of research which may also be of interest. Not including this suggestion, should not impede the publication of this protocol which is clearly acceptable for publication as it stands now.
--

VERSION 1 – AUTHOR RESPONSE

Reviewer 1

- I do miss an overview of when the study was initiated/ data collection started, in both the original intervention trial, but also in the mixed methods trial, and the expected time of finalizing data collection.

A study timeline section has been added at the end of the main document before the references.

STUDY TIMELINES

Funding for this study was approved on the 14th August 2012

The Trial and qualitative evaluation and process evaluation all started 1st September 2013 and finished 30th November 2018.

- The authors have included questionnaires about bowel health and pelvic organ prolapse which I believe to be an important aspect. Thus, I would also like to enquire about whether the authors plan to include any questions in the planned interviews about other pelvic floor disorders (PFD), and how experiencing multiple PFDs may influence adherence and the participants' experience of PFMT and the individual follow up.

The authors did not specifically ask about other pelvic floor disorders but did explore barriers and facilitators more generally.

Reviewer 2

- Tables in pages 34/40 and 35/40 did not appear correctly (went off page and over text).

The authors cannot address this within the main document as this is a formatting issue which occurred with the upload process. The authors will check all tables appear correctly within the final manuscript.

- Spirit Checklist 32 A model consent form does not seem to have been included as an appendix

The Spirit checklist model consent form was uploaded with the trial protocol rather than with the process evaluation and qualitative study. We cite the trial protocol.

Reviewer 3

- My only observation is that there may also be an opportunity to administer some validated patient reported outcomes which assess experiences with treatment, these could add some quantitative data to this strand of research which may also be of interest.

The authors thank this reviewer for his comments however, given the large volume of data already being collected this is beyond the resources of this study but will be considered as a limitation when we publish the results.